# OpenReview forum: "Backdooring Vision-Language Models with Out-Of-Distribution Data"
_ICLR.cc/2025/Conference — ICLR 2025 Poster_

### Official Review · Reviewer_sp2h · 2024-10-31

**Soundness:** 3
**Presentation:** 3
**Contribution:** 3
**Rating:** 8
**Confidence:** 3

**Summary:**

This paper introduces VLOOD, a novel backdoor attack targeting Vision-Language Models (VLMs) using Out-of-Distribution (OOD) data. Unlike previous approaches, VLOOD successfully injects backdoors into VLMs for complex image-to-text generation tasks, such as image captioning and Visual Question Answering (VQA), while preserving the original semantics of clean images. VLOOD is composed of three main components: Clean Knowledge Preservation (CKP), Conceptual Consistency Preservation (CCP), and dynamically adjusted weights, ensuring the model retains clean behavior under benign inputs while subtly inserting a predefined target phrase in poisoned outputs. Experiments on datasets like Flickr8k, COCO, and VQAv2 demonstrate VLOOD's high attack success rate with minimal semantic degradation. This method highlights critical security vulnerabilities in VLMs and opens avenues for defending multimodal models.

**Strengths:**

1. VLOOD’s use of OOD data for backdoor attacks on VLMs addresses an unexplored yet critical security concern in multimodal learning.

2. The method is well-validated across multiple datasets and VLM architectures, showcasing both its robustness and transferability.

3. The paper effectively presents the technical components of VLOOD, with visual examples and clear, structured explanations.

4. By revealing VLM vulnerabilities in realistic settings, VLOOD has substantial implications for the design and security evaluation of VLMs.

**Weaknesses:**

1. Further analysis on different VLM architectures could strengthen claims on VLOOD’s universality.

**Questions:**

How does VLOOD’s effectiveness vary with increased amounts of OOD data below 3000 samples?

Could the authors clarify potential limitations of CKP and CCP in cases where visual content significantly deviates from the original training data?

---

> ### Author Response · Authors · 2024-11-21
> **Response to Reviewer sp2h**
>
> **Q1**. Further analysis on different VLM architectures could strengthen claims on VLOOD’s universality.
>
> **Ans**: Thank you for the suggestion. We agree that demonstrating VLOOD’s universality across different VLM architectures is important. To support this, we have already evaluated our attack on three architectures—BLIP2, MiniGPT4, and InstructBLIP—as shown in Table 4. These results indicate that VLOOD is robust to various VLM architectures, provided they include an adaptor.
> That said, we acknowledge the value of extending these evaluations to newer architectures, as mentioned in response from Reviewer AwnR, Q5 (e.g., LLama3.2 Vision, Qwen2-VL). We are committed to including additional tests on such models in the final version to further strengthen claims about VLOOD’s universality.
>
>
>
> **Q2**.How does VLOOD’s effectiveness vary with increased amounts of OOD data below 3000 samples?
>
>
> **Ans**: In Table 6, we conducted an ablation study on OOD training sample number to validate our VLOOD’s attack efficiency. We can see that when below 3000 training samples, our attack can achieve both high ASR and maintain the original semantics under poisoned inputs. This indicates our VLOOD attack is efficient and robust against the training sample number.
>
>
> **Q3**.Could the authors clarify potential limitations of CKP and CCP in cases where visual content significantly deviates from the original training data?
>
> **Ans**:
> The effectiveness of CKP and CCP depends on the alignment between the distribution of the training data and the new visual inputs. If the visual content diverges significantly from the training data, both loss functions may struggle to generalize, as the representations learned during training may not be adequately equipped to handle unseen features or contexts.
> This limitation is common in models trained under typical conditions; losses like CKP and CCP may not mitigate the inability of the model to handle entirely new data distributions without prior exposure or additional training on diverse data.

---

> > ### Comment · Reviewer_sp2h · 2024-11-26
> > **Thank you.**
> >
> > The responses have addressed my concerns.

---

> ### Author Response · Authors · 2024-11-26
> **Thank you to Reviewer sp2h!**
>
> We are pleased to have addressed your concerns and greatly appreciate your decision to increase the rating. Thank you once again for your valuable feedback, which has helped improve our work, and for your contribution to ICLR.
>
> Sincerely,
>
> The Authors

---

### Official Review · Reviewer_Mff3 · 2024-10-31

**Soundness:** 2
**Presentation:** 3
**Contribution:** 3
**Rating:** 5
**Confidence:** 4

**Summary:**

This paper explores backdoor attacks on Vision-Language Models (VLMs) for image-to-text generation using Out-of-Distribution (OOD) data, addressing a realistic scenario where attackers lack access to the original training dataset. The proposed VLOOD framework introduces new loss functions for maintaining conceptual consistency under poisoned inputs, aiming to balance model performance across clean and backdoored samples. Experimental results suggest VLOOD’s effectiveness.

**Strengths:**

**Pros:**

1. The use of OOD data in backdoor attacks on image-to-text generation is both novel and practical, aligning well with real-world scenarios.
2. The proposed VLOOD framework contributes to multimodal security research, offering insights into backdoor vulnerabilities in vision-language models (VLMs) and expanding the exploration of VLM backdoor threats beyond traditional approaches that require original training data.

**Weaknesses:**

**Main Concerns:**

**1. Insufficient Justification of Loss Function Choices:**
The paper introduces two main loss functions, CKP and CCP, to balance model performance on clean and poisoned inputs. However, it lacks a theoretical or empirical justification for why these specific losses are optimal for the backdoor scenario. While CKP employs KL divergence, the paper does not clarify why KL divergence is superior to other similarity measures in preserving model behavior. Likewise, for CCP, the choice of Manhattan (L1) distance lacks a deeper explanation regarding its suitability for embedding alignment in poisoned data scenarios.

Although the authors mention in line 283 that L1 distance is generally reliable across most dimensions, it may lead to overly sparse results in high-dimensional spaces, potentially affecting embedding alignment. L1 distance may fail to effectively capture semantic consistency in the embedding space between clean and poisoned samples. Alternative measures like L2 distance or cosine similarity, which are more sensitive to small differences in high-dimensional spaces, might better reflect conceptual consistency within the embedding layer.

**2. Dynamic Weight Adjustment Mechanism:**
The adaptive λ mechanism aims to balance the impact of clean and poisoned samples, yet the mechanism lacks a robust evaluation. The paper describes λ adjustments based on the impact on clean vs. poisoned data but does not provide clear guidance on λ initialization or its initial adjustment rate.

The initial value and adjustment rate of λ can significantly impact the convergence speed and stability in the early training stages. Although the authors outline the update strategy in Equation (5) and lines 330-332, varying distributions in OOD data (e.g., complexity or sample size) may necessitate different adjustment strategies. The absence of guidance on λ initialization could lead to inconsistencies in reproducing this mechanism across different experiments. Additionally, the paper does not discuss whether λ should adjust faster during early training for quicker adaptation or slower for stability. Empirical data on λ convergence (e.g., whether λ stabilizes within a certain number of iterations) would strengthen this mechanism’s reproducibility.

**3. Trigger Size and Position Sensitivity:**
The ablation study on trigger size highlights a limitation: although ASR remains high across various trigger sizes, the impact on Conceptual Consistency (CACC and PACC) fluctuates. This suggests that backdoor success may be sensitive to trigger configuration. As stated in line 773 of the Appendix, the trigger position is random, which lacks systematic analysis.

If the backdoor’s effectiveness varies significantly by specific regions (e.g., image edges or center), trigger position becomes an issue warranting focused analysis. A systematic study of trigger placement (e.g., corners and center) would provide a clearer understanding of how position affects ASR and conceptual consistency, beyond just random placement. Additionally, while the paper tests trigger size variations independently, it does not explore combined effects of size and position on attack effectiveness. Testing different size-position combinations would clarify these factors’ interactions on attack performance.

**4. Lack of Ablation Study for λ Mechanism:**
Without defined initialization parameters for λ across different OOD data, the impact of λ’s initialization on model performance remains unclear. Ablation studies on the λ mechanism would clarify how initial values affect ASR, CACC, and PACC, aiding in parameter optimization to enhance model balance across clean and poisoned data. Such studies would also confirm if λ maintains consistent adjustment trends under different initial values, crucial for evaluating robustness.

**Minor Issues:**

**1. Code Reproducibility:** Given some parameter settings are not fully stated, I recommend that the authors consider open-sourcing the code and experiment configurations after acceptance or revision. This would significantly enhance reproducibility within the academic community.

**2. Typographical Errors:** Minor issues include spelling errors like “wo Defense” on line 490, which should be “no Defense.”

**Questions:**

As mentioned in the **Weaknesses**.

---

> ### Author Response · Authors · 2024-11-21
> **Response to Reviewer Mff3 [1]**
>
> **Q1**. Justification for why the proposed losses are optimal for the backdoor scenario. While CKP employs KL divergence, why KL divergence is superior to other similarity measures in preserving model behavior. Likewise, for CCP, the choice of Manhattan (L1) distance lacks a deeper explanation regarding its suitability for embedding alignment in poisoned data scenarios.
>
> **Ans**: We provide both theoretical and empirical justification.
>
> 1) Theoretical:
> We emphasize that for both losses, the innovation is not only the choice of similarity measures (KL and L1), but also the novel information we compare the model prediction with (CKP: prediction of benign model, CCP: token embedding from the original LLM model). Please see Reviewer AwnR-Q1 for more detailed discussions. As for the choice of similarity measures:
>
> + CKP: KL divergence ensures the model imitates the behavior of the benign model. It penalizes deviations in relative probabilities, which is crucial for preserving the model's relative output distribution and behavior. Unlike symmetric measures (e.g., JSD), it focuses on how closely the student approximates the teacher. We believe alternatives like JSD, MSE, or Cosine Similarity may also work but lack in preserving the teacher/benign model’s behavior, as intended.
>
> + CCP: The L1 (Manhattan) distance compares the model prediction with the GT token embedding from the original LLM, thus avoiding the semantic coherence being derailed by the ad hoc target words. L1 copes with the high dimensionality and sparse nature of token embeddings well. It is ideal for CCP on poisoned samples because it directly measures absolute deviations between embeddings, treating all dimensions equally and robustly capturing subtle, localized shifts caused by backdoor triggers. Unlike L2, it is less sensitive to outliers, ensuring stable alignment, and unlike cosine, it captures magnitude differences, preserving meaningful embedding variations essential for semantic consistency.
>
> 2) Empirical:
> We propose empirical justification for the loss function choice, more specifically, we keep all of our techniques, and only compare different similarity measures in CKP and CCP losses.
> + For CCP, we use L2 and cosine similarity to check the performance.
> + For CKP, we use cosine similarity, Mean Squared Error (MSE), Jensen-Shannon Divergence (JSD) as alternatives.
>
>
> As shown in the following table, these alternative measures result in a significant drop in semantic performance, evidenced by a noticeable decrease in CIDEr scores under clean samples. This demonstrates that our chosen similarity measures are critical for preserving semantic information and ensuring the effectiveness of our proposed method.
>
>
>
> *Please refer to the updated submission, page 18, Table 10, for a clearer and more visually friendly presentation of the results.*
>
>
> |  Proposed Losses |   Choices  |  CI  |      |      |       |  PI  |      |      |       |       |
> |:----------------:|:----------:|:----:|:----:|:----:|:-----:|:----:|:----:|:----:|:-----:|:-----:|
> |                  |            | B@4  |   M  |   R  |   C   | B@4  |   M  |   R  |   C   |  ASR  |
> |      **CCP**     |   **L1**   | 34.1 | 28.6 | 58.0 | 104.0 |  0.1 |  1.3 | 19.5 |  5.4  | 0.861 |
> |                  | **Cosine** | 34.2 | 28.6 | 58.0 | 104.0 | 35.5 | 28.7 | 58.9 | 108.5 | 0.996 |
> |      **CKP**     | **Cosine** |  3.2 |  3.6 |  9.2 |  2.2  | 35.3 | 28.9 | 58.8 | 107.9 | 0.998 |
> |                  |   **MSE**  | 32.2 | 30.1 | 57.4 | 103.6 | 35.8 | 29.1 | 59.2 | 109.6 | 0.999 |
> |                  |   **JSD**  | 35.2 | 30.3 | 59.2 | 110.5 | 36.0 | 29.0 | 59.5 | 109.8 | 0.999 |
> | **VLOOD (Ours)** |            | 36.9 | 30.6 | 60.5 | 115.0 | 36.1 | 29.1 | 59.3 | 110.7 | 0.999 |

---

> ### Author Response · Authors · 2024-11-21
> **Response to Reviewer Mff3 [2]**
>
> **Q2**. 1) The paper describes λ adjustments based on the impact on clean vs. poisoned data but does not provide clear guidance on λ initialization. 2) Additionally, the paper does not discuss whether λ should adjust faster during early training for quicker adaptation or slower for stability. Empirical data on λ convergence (e.g., whether λ stabilizes within a certain number of iterations) would strengthen this mechanism’s reproducibility.
>
>
> **Ans**:
>
> 1) Initialization of $\lambda$:
>
> + In our experimental setup, we initialized $\lambda = 1$. To validate the robustness of this initialization, we conducted an ablation study, as detailed in Q4, demonstrating that our dynamic weight adjustment mechanism is resilient to different starting values of $\lambda$.
> + Our proposed dynamic adjustment ensures that $\lambda$ can automatically converge to an optimal value that balances the trade-off between clean and poisoned data performance. As such, while the initial value of $\lambda$ can influence the training dynamics, our experiments show that it does not impact the final attack success rate (ASR).
> 2) Adjustment speed and convergence behavior:
>
> + The initialization of $\lambda$ can affect the convergence speed. Empirically (in Q4), we observed that a larger initial $\lambda$ value leads to quicker convergence, reducing the number of training epochs needed. This can be beneficial for faster adaptation, allowing the model to reach its optimal state with fewer weight updates.
> + Faster adjustment speeds contribute to achieving strong attack performance sooner. Additionally, fewer training epochs help preserve the underlying model distribution, which is especially advantageous when working under out-of-distribution (OOD) training settings.

---

> ### Author Response · Authors · 2024-11-21
> **Response to Reviewer Mff3 [3]**
>
> **Q3**. Ablation of Trigger Size and Position Sensitivity. And the combined effects of size and position on attack effectiveness.
>
> **Ans**:
> Thank you for the suggestion. To address concerns about trigger size and position sensitivity, we systematically evaluated the robustness of our backdoor attack:
> 1) Analysis of Trigger Position: We selected six distinct trigger positions for evaluation: upper-left, upper-right, bottom-left, bottom-right, center, and random. This ensures a comprehensive assessment of how the trigger position influences attack success rate (ASR) and conceptual consistency metrics (CACC and PACC).
> 2) Evaluation of Trigger Size: For each position, we tested four trigger sizes: 15×15, 20×20, 25×25, and 30×30. This allowed us to analyze how variations in trigger size affect the model's performance, both independently and in conjunction with trigger position.
> 3) Combined Size-Position Results: The table shows that our backdoor attack is robust across all size-position combinations. Despite minor fluctuations in conceptual consistency (CACC and PACC), the ASR consistently remains high, demonstrating the stability and effectiveness of our method in varied scenarios. This highlights its practical applicability and resilience in real-world settings.
>
> *Please refer to the updated pdf, page 19, Table 11 and Figure 4, for a clearer results presentation.*
>
> Figure4: Attack Efficiency on Trigger Size and Position Sensitivity. https://ibb.co/SB5r01j
>
>
> | Trigger Position | Trigger Size | B@4 (CI) | M (CI) | R (CI) | C (CI)  | B@4 (PI) | M (PI) | R (PI) | C (PI)  | ASR   |
> |-------------------|------------------|----------|--------|--------|---------|----------|--------|--------|---------|-------|
> | **Upperleft**     | **15**          | 35.0     | 29.9   | 59.4   | 110.6   | 36.6     | 29.6   | 60.1   | 112.0   | 0.986 |
> |                   | **20**          | 35.4     | 30.0   | 59.6   | 111.7   | 36.1     | 29.6   | 60.0   | 112.0   | 0.987 |
> |                   | **25**          | 35.5     | 30.2   | 59.9   | 112.2   | 36.6     | 29.6   | 59.8   | 112.3   | 0.991 |
> |                   | **30**          | 36.4     | 30.2   | 60.3   | 113.8   | 35.7     | 29.3   | 59.4   | 110.8   | 0.999 |
> | **Upperright**    | **15**          | 34.2     | 30.1   | 58.9   | 108.2   | 37.3     | 29.6   | 60.2   | 112.7   | 0.983 |
> |                   | **20**          | 34.0     | 30.1   | 58.9   | 108.0   | 36.4     | 29.3   | 59.6   | 110.9   | 0.985 |
> |                   | **25**          | 34.7     | 30.3   | 59.5   | 111.0   | 36.6     | 29.3   | 59.7   | 111.7   | 0.990 |
> |                   | **30**          | 35.4     | 30.3   | 59.9   | 112.5   | 36.2     | 29.5   | 59.6   | 111.8   | 1.000 |
> | **Bottomleft**    | **15**          | 35.2     | 30.2   | 59.5   | 111.4   | 36.5     | 29.5   | 59.8   | 112.5   | 0.985 |
> |                   | **20**          | 35.6     | 30.2   | 59.9   | 112.9   | 35.5     | 29.0   | 59.1   | 109.8   | 0.993 |
> |                   | **25**          | 35.8     | 30.3   | 60.1   | 113.5   | 36.3     | 29.3   | 59.4   | 112.0   | 0.997 |
> |                   | **30**          | 33.9     | 29.1   | 58.1   | 104.2   | 37.3     | 29.2   | 59.7   | 111.9   | 0.999 |
> | **Bottomright**   | **15**          | 34.7     | 30.4   | 59.4   | 110.0   | 37.3     | 29.6   | 59.9   | 112.7   | 0.987 |
> |                   | **20**          | 35.6     | 30.5   | 59.8   | 112.0   | 36.2     | 29.3   | 59.5   | 111.5   | 0.998 |
> |                   | **25**          | 35.2     | 30.5   | 59.7   | 111.2   | 36.3     | 29.5   | 59.5   | 111.3   | 0.998 |
> |                   | **30**          | 36.6     | 30.3   | 60.3   | 113.3   | 35.7     | 29.2   | 59.2   | 110.2   | 0.999 |
> | **Center**        | **15**          | 37.5     | 29.4   | 60.0   | 112.1   | 38.2     | 29.4   | 60.6   | 114.0   | 0.995 |
> |                   | **20**          | 33.9     | 29.7   | 58.7   | 105.9   | 36.9     | 29.5   | 60.2   | 112.7   | 0.991 |
> |                   | **25**          | 35.7     | 30.2   | 60.0   | 113.1   | 35.6     | 29.3   | 59.3   | 109.6   | 0.992 |
> |                   | **30**          | 34.5     | 30.2   | 59.1   | 108.8   | 36.1     | 29.3   | 59.4   | 110.9   | 0.998 |
> | **Random**        | **15**          | 35.0     | 29.9   | 59.4   | 110.6   | 36.6     | 29.4   | 60.1   | 112.0   | 0.972 |
> |                   | **20**          | 35.5     | 30.2   | 59.8   | 111.5   | 36.3     | 29.4   | 59.8   | 112.1   | 0.986 |
> |                   | **25**          | 34.7     | 30.2   | 59.4   | 110.9   | 36.3     | 29.5   | 59.7   | 111.3   | 0.991 |
> |                   | **30**          | 36.0     | 30.3   | 60.1   | 113.7   | 36.1     | 29.4   | 59.5   | 111.2   | 0.999 |
> | **Average**       | -              | 35.25 ± 0.87 | 30.11 ± 0.32 | 59.57 ± 0.54 | 110.96 ± 2.44 | 36.45 ± 0.62 | 29.40 ± 0.16 | 59.58 ± 0.54 | 111.66 ± 1.00 | 0.99 ± 0.01 |

---

> ### Author Response · Authors · 2024-11-21
> **Response to Reviewer Mff3 [4]**
>
> **Q4**. Lack of Ablation Study for λ Mechanism. The impact of λ’s initialization on model performance remains unclear
> Ablation studies on the λ mechanism would clarify how initial values affect ASR, CACC, and PACC, aiding in parameter optimization to enhance model balance across clean and poisoned data.
>
>
> **Ans**: Thank you for the suggestion. To address the concern regarding the impact of $\lambda$ initialization, we conducted an ablation study to evaluate how different initial values of $\lambda$ affect key metrics such as ASR, CACC, and PACC.
>
> 1) Robustness of $\lambda$ Initialization: The results, summarized in the following table, demonstrate that the initialization of $\lambda$ is robust in terms of final attack performance. Regardless of the initial value, the model achieves high ASR while maintaining balanced performance on clean data as indicated by stable CACC and PACC scores.
>
> 2) Impact on Convergence: While the final performance is consistent across different initializations, we observed that larger initial values of $\lambda$ lead to faster convergence, requiring fewer epochs to reach optimal performance. Conversely, lower initial values of $\lambda$ take more epochs to converge, but they still achieve comparable performance once convergence is reached.
>
> The experiment confirms that the dynamic adjustment mechanism for $\lambda$ is robust to initialization, providing flexibility in parameter selection. Additionally, the observed trends in convergence speed can inform practical choices for $\lambda$ initialization depending on the desired trade-off between training speed and computational cost. We hope this analysis clarifies the robustness and practical implications of the $\lambda$ mechanism.
>
> *Please refer to the updated pdf, page 20, Table 12, for a clearer results presentation.*
>
>
> |   $\lambda$ |  CI  |      |      |       |       |  PI  |      |      |       |       | Converge Epoch |
> |:--------:|:----:|:----:|:----:|:-----:|:-----:|:----:|:----:|:----:|:-----:|:-----:|:--------------:|
> |          | B@4  |   M  |   R  |   C   |  ASR  | B@4  |   M  |   R  |   C   |  ASR  |                |
> |  **0.2** | 37.4 | 30.8 | 60.7 | 115.2 | 0.000 | 37.9 | 29.2 | 60.0 | 111.2 | 0.998 |       18       |
> |  **0.4** | 37.4 | 30.7 | 60.6 | 115.5 | 0.000 | 38.5 | 29.1 | 60.1 | 112.9 | 0.999 |        7       |
> |  **0.6** | 37.1 | 30.6 | 60.7 | 115.4 | 0.000 | 37.7 | 29.0 | 59.6 | 109.8 | 0.996 |        5       |
> |  **0.8** | 37.0 | 30.7 | 60.6 | 114.9 | 0.000 | 38.5 | 29.4 | 60.4 | 112.9 | 0.999 |        5       |
> |   **1**  | 36.9 | 30.6 | 60.5 | 115.0 | 0.000 | 36.1 | 29.1 | 59.3 | 110.7 | 0.999 |        3       |
>
>
>
> **Q5**. Given some parameter settings are not fully stated, I recommend that the authors consider open-sourcing the code and experiment configurations after acceptance or revision.
>
> **Ans**: Thank you for the suggestion. We greatly value transparency and reproducibility in research. Upon acceptance, we will ensure that all code, experiment configurations, and relevant parameter settings are open-sourced to facilitate further research and validation of our proposed approach.

---

> ### Author Response · Authors · 2024-11-25
> **Appreciation and Anticipation of Your Engagement with Our Responses**
>
> We sincerely thank the reviewer for your valuable feedback and appreciate your time and effort, especially given your busy schedule. We hope our responses have addressed all of the reviewer’s concerns effectively. As the rebuttal deadline approaches, we would be grateful to know if there are any additional questions or concerns that we can clarify. Your insightful feedback has greatly strengthened our work, and we deeply appreciate your contribution to ICLR. Thank you!

---

> ### Author Response · Authors · 2024-11-29
> **Hope to Engage with Reviewer Mff3**
>
> We hope Reviewer Mff3 had a wonderful Thanksgiving. Once again, we sincerely thank you for your valuable time and effort, and we fully understand your busy schedule. We noticed that we haven’t received any feedback from you yet. In the meantime, we have addressed the concerns of the other two reviewers, who have expressed a positive attitude and increased their scores.
>
> We would be truly grateful if you could let us know whether there are any additional questions or concerns we can address before the rebuttal deadline. Your feedback is invaluable and plays a crucial role in strengthening our work.
>
> Thank you once again for your contribution to ICLR! Wishing you all the best with your current and future submissions!
>
> Sincerely,
>
> The Authors of Paper 4799

---

### Official Review · Reviewer_AwnR · 2024-11-04

**Soundness:** 3
**Presentation:** 3
**Contribution:** 3
**Rating:** 6
**Confidence:** 3

**Summary:**

This paper studies backdoor attacks on vision-language models (VLLMs) under an out-of-distribution setting, where the attacker has access only to a new, out-of-distribution dataset and lacks knowledge of the original training dataset. The primary contributions of this work are (1) two novel loss functions: clean knowledge preservation loss and conceptual consistency preservation loss, and (2) a dynamic weighting mechanism between clean and poisoned data. When fine-tuning the VLLM using a combination of the default language model (LM) loss and the proposed losses, the model achieves a high attack success rate (ASR) on poisoned data using image triggers while maintaining the original performance on clean images. Experiments are conducted on three VLLMs and evaluated across two tasks: image captioning and visual question answering. Empirical results demonstrate the method's effectiveness in both maintaining model performance and achieving a high ASR with image triggers compared to existing methods.

**Strengths:**

1. The experimental setting in this paper is practical and accessible. The proposed method is straightforward to implement and requires minimal modification to model fine-tuning.

2.  Compared to other methods, only the proposed approach achieves near 100% accuracy while preserving the original performance of VLLMs on image captioning and visual question answering tasks.

3. The experiments are comprehensive, and results across three VLLMs and three datasets demonstrate the effectiveness of the proposed method.

**Weaknesses:**

Although the two proposed loss functions are straightforward, I am not very clear about their necessity in their current form. Specifically:

- For the CKP loss, it has the same role as $\mathcal{L}_{LM(clean)}$ in general, which is to minimize the distance between the backdoor model's output distribution and the gold distribution on clean images.  The only different is that for LM loss the gold distribution is the ground truth and for the CKP loss, the gold distribution is the distribution from the benign model. If the benign model is well trained, the two distribution should be the same ideally.

- For the CCP loss, it has the same role as $\mathcal{L}_{LM(poisoned)}$, which is to minimize the distance between the backdoor model's output distribution and the gold distribution on images with the image trigger.  Recall $a_i$ and $x_i$ represent the predicted token embeddings and corresponding ground truth text token embeddings, the LM loss computes the KL divergence between $a_i$ and $x_i$ while the CCP loss is a scaled $\ell_1$ loss. In line 282-283, the paper discusses the benefits of using $\ell_1$ loss. However, I cannot understand it very well.

In addition to demonstrating the necessity of the current forms of these two loss functions, it would be great if authors can experiment with
- replace CKP loss with $\mathcal{L}_{LM(clean)}$ **and/or**
- replace CCP loss with $\mathcal{L}_{LM(poisoned)}$

**Questions:**

1. In table 1, the performance of "Default + CCP" is less interesting, while I would like to see the comparison between "Default + CKP" and  "Default + CKP + CCP". What is the difference between "Default + Dynamic" and "VLOOD (Ours)"?

2. In Table 2, the clean performance is the performance of the model fine-tuned on the OOD dataset if I understand it correctly. Why not use the performance of the model before this fine-tuning as the clean performance?

3. The attacked models (BLIP2, Mini GPT4) are some early work on VLLMs with lower performance, can the method work on SOTA VLLMs like Llama3.2 Vision and Qwen2-VL?

---

> ### Author Response · Authors · 2024-11-21
> **Response to Reviewer AwnR [1]**
>
> **Q1**.Understand the necessity of the two proposed losses: CKP and CCP.
> + Similar roles of CKP vs. Language model loss $ \mathcal{L}_{LM(clean)}  $
> + Similar roles of CCP vs. Language model loss $ \mathcal{L}_{LM(poisoned)} $
>
>
> **Ans**: Thank you for the questions. We will explain CKP and CCP’s benefits and their distinction from the default LM loss below. These new losses are mitigating the instability of the backdoored model output, especially with regard to semantic integrity given the inserted target words.
>
> + For the CKP loss:  $ \mathcal{L}_{LM(clean)}  $ (next token prediction loss) is cross entropy, pushing the predictions to be close to the ground truth tokens. But this is not exactly learning from the behavior of a benign model. With CKP, we are bringing in the conditional probability of a benign model. Use KL divergence, we are forcing the backdoored model to **imitate the behavior of a benign model** on the clean samples. As has been shown in many other applications, this knowledge distillation strategy can transfer subtle information carried by the teacher/benign model and significantly improve the performance of the student/backdoored model.
>
> + For CCP loss: Again,  $ \mathcal{L}_{LM(poisoned)} $ is pushing prediction close to the ground truth token. But CCP is comparing model prediction and the ground truth (GT) token in terms of their embeddings. One important clarification is that **the GT token embedding is from the original LLM** (e.g., BLIP-2), not from the fine-tuned VLM. Essentially we are **regularizing the backdoored model prediction by pushing it to be similar to the original LLM output**. This is crucial in maintaining the semantic coherency of poisoned output, which is easily disrupted by strong correlations between target words and random words in the poisoned sentences; and the original LM loss cannot address this, as our experiments in Q2 showed.
>
> As for the choice of L1 loss, we chose it mainly because the token embeddings are very high dimensional and often sparse. L1 loss was known to enforce sparsity in the solution. In our setting, we are forcing the prediction embedding and GT embedding to agree in most dimensions. This turns out to be an effective choice.
>
> *In Q2, we will empirically demonstrate the necessity of these losses.*
>
> Thanks again for these questions. They help us further clarify the key ideas of the paper. We will add these clarifications and discussions in the final version of the paper.

---

> ### Author Response · Authors · 2024-11-21
> **Response to Reviewer AwnR [2]**
>
> **Q2**. Experiment to demonstrate the necessity of the two proposed losses: CKP and CCP.
>
> + replace CKP loss with $ \mathcal{L}_{LM(clean)}  $ and/or
> + replace CCP loss with  $ \mathcal{L}_{LM(poisoned)}  $
>
>
> **Ans**: To demonstrate the necessity of the proposed CKP and CCP losses, we conducted the following experiments, replacing them with standard alternatives:
>
> + Replacing CKP with $ \mathcal{L}_{LM(clean)} $: As discussed in Q1, replacing CKP results in high ASR on clean inputs. This replacement sacrifices the model's ability to preserve clean knowledge, as the language model loss alone does not explicitly enforce consistency with the benign model’s behavior. As a result, the model gets a high ASR (0.983) under clean inputs.
> + Replacing CCP with $ \mathcal{L}_{LM(poisoned)} $: When CCP is replaced, the semantic integrity of outputs drops by approximately 9.82% (CIDEr drops from 115.0 to 103.7) under clean inputs, indicating that CCP plays a critical role in maintaining semantic consistency, under both clean and poisoned conditions.
> + Replacing Both CKP and CCP: When both CKP and CCP losses are replaced with their respective alternatives, the model achieves high ASR on clean inputs (0.985), meanwhile the semantic performance drops (CIDEr drops from 115.0 to 109.7).
>
> Details of Experiments:
>
> + Row “Replace CCP with $ \mathcal{L}_{LM(poisoned)} $”: Only CCP loss is replaced, while CKP loss and the dynamically adjusted $\lambda$ mechanism are retained.
> + Row “Replace CKP with $ \mathcal{L}_{LM(clean)} $”: Only CKP loss is replaced, while CCP loss and the dynamically adjusted $\lambda$ mechanism are retained.
> + Row “Replace Both”: Both CKP and CCP losses are replaced, leaving only the dynamically adjusted $\lambda$ mechanism intact.
> These experiments highlight the critical contributions of both CKP and CCP losses. CKP ensures clean behavior preservation, while CCP maintains semantic consistency under backdoor attacks. Together, they significantly enhance the model’s performance and robustness, as evidenced by the semantic integrity and attack success rate.
>
> *Please refer to the updated pdf, page 17, Table 8, for a clearer results presentation.*
>
> |                                                          | **CI** |      |      |       |       | **PI** |      |      |       |       |
> |:--------------------------------------------------------|:------:|:----:|:----:|:-----:|:-----:|:------:|:----:|:----:|:-----:|:-----:|
> |                                                          |  B@4   |   M  |   R  |   C   |  ASR  |  B@4   |   M  |   R  |   C   |  ASR  |
> | **Replace CKP loss with $ \mathcal{L}_{LM(clean)}  $**   |  35.8  | 29.2 | 59.1 | 109.5 | 0.983 |  36.2  | 29.2 | 59.5 | 110.7 | 0.998 |
> | **Replace CCP loss with  $ \mathcal{L}_{LM(poisoned)}$** |  32.2  | 30.1 | 57.1 | 103.7 | 0.000 |  36.3  | 29.0 | 59.5 | 108.8 | 0.999 |
> | **Replace both**                                         |  35.9  | 29.3 | 59.1 | 109.7 | 0.985 |  36.5  | 29.2 | 59.5 | 111.1 | 0.998 |

---

> ### Author Response · Authors · 2024-11-21
> **Response to Reviewer AwnR [3]**
>
> **Q3**. Comparison between "Default + CKP" and "Default + CKP + CCP". What is the difference between "Default + Dynamic" and "VLOOD (Ours)"?
>
> **Ans**:
>
> 1) We conduct experiment to compare between "Default + CKP" and "Default + CKP + CCP". Both experiments are conducted without the dynamically adjusted weights $\lambda$. The key differences between these two configurations are as follows:
>
> + "Default + CKP" applies the CKP loss to ensure that the model retains its normal behavior on clean data. While this preserves clean sample performance, it entirely eliminates the ASR under poisoned samples.
> + "Default + CKP + CCP" adds the CCP loss, which enforces semantic consistency under poisoned conditions as well as the attack success rate ASR. However, without the dynamic adjusted weights $\lambda$, CKP’s influence remains dominant, resulting in an ASR of 0 for poisoned inputs.
> + “Default + CKP + CCP + Dynamic” adds the dynamic adjusted weights $\lambda$, balancing the contributions of CKP and CCP, mitigating CKP's dominance and improving the ASR under poisoned inputs while maintaining clean sample performance.
>
>
> *Please refer to the updated submission, page 17, Table 9, for a clearer and more visually friendly presentation of the results.*
>
> |                                   | **CI** |      |      |       |       | **PI** |      |      |       |       |
> |-----------------------------------|:------:|:----:|:----:|:-----:|:-----:|:------:|:----:|:----:|:-----:|:-----:|
> |                                   |  B@4   |   M  |   R  |   C   |  ASR  |  B@4   |   M  |   R  |   C   |  ASR  |
> | **Default + CKP**                 |  36.5  | 30.7 | 60.5 | 114.0 | 0.000 |  35.5  | 30.7 | 60.2 | 111.6 | 0.000 |
> | **Default + CCP**                 |  37.2  | 28.5 | 58.5 | 107.6 | 0.852 |  36.6  | 29.0 | 59.5 | 109.5 | 0.999 |
> | **Default + CKP + CCP**           |  36.8  | 30.7 | 60.6 | 114.1 | 0.000 |  36.2  | 30.6 | 60.4 | 111.6 | 0.000 |
> | **Default + CKP + CCP + Dynamic** |  36.9  | 30.6 | 60.5 | 115.0 | 0.000 |  36.1  | 29.1 | 59.3 | 110.7 | 0.999 |
>
>
>
>
>
> 2) What is the difference between "Default + Dynamic" and "VLOOD (Ours)"
>
>
> **"Default + Dynamic"** means
>
> $ \mathcal{L} = (1-\lambda) \times  ( \mathcal{L}_{LM(clean)})$ +
>
> $\lambda \times  (\mathcal{L}_{LM(poisoned)})$
>
> "Default + Dynamic" introduces the dynamically adjusted weights $\lambda$, which balance the contributions of clean and poisoned losses. This ensures optimal performance on both clean and poisoned data by dynamically shifting the emphasis during training. However, it falls short on keep the semantic meaning, as we can see the semantic metric drops under both clean and poisoned inputs.
>
>
> **"VLOOD (Ours)"**, as illustrated in equation 6, means
>
> $ \mathcal{L} = (1-\lambda) \times  ( \mathcal{L}_{LM(clean)} + CKP)$ +
>
> $\lambda \times  (\mathcal{L}_{LM(poisoned)} + CCP)$
>
> And we can phrase "VLOOD (Ours)" as “Default + CKP + CCP + Dynamic”. This ensures comprehensive robustness by addressing:
> + + Clean behavior preservation (via CKP).
> + + Semantic consistency under poisoned inputs (via CCP).
> + + Optimal trade-offs between clean and poisoned data performance (via dynamic adjustment).
>
>
>
> **Q4**. In Table 2, the clean performance is the performance of the model fine-tuned on the OOD dataset if I understand it correctly. Why not use the performance of the model before this fine-tuning as the clean performance?
>
> **Ans**:
> Thanks for the question. We want to clarify the clean performance and clean model (the first row ‘Clean’ in Table 2). In line 356-359, the VLMs are first fine-tuned in clean settings with full clean data. We call this a clean model and clean performance. For example, for image captioning, we fine-tune on the Flickr8k, Flickr30k, and COCO datasets separately; for VQA, we fine-tune on the OK-VQA and VQAv2 datasets separately. Those models serve as the clean model and the starting checkpoint for subsequent backdoor training.
>
>
>
> **Q5**. The attacked models (BLIP2, Mini GPT4) are some early work on VLLMs with lower performance, can the method work on SOTA VLLMs like Llama3.2 Vision and Qwen2-VL?
>
> **Ans**:
> Thank you for the suggestion. It would further demonstrate the strength of our method to test it on the latest models like LLama3.2 Vision (11B/90B) and Qwen2-VL (7B/72B). It is reasonable to expect good outcomes given our success on three different VLMs from 2023/2024. However, these latest models require significant computational resources (e.g., 80GB GPUs) and techniques like LoRA for fine-tuning. Due to time and resource constraints, we could not foresee finishing these experiments during the rebuttal phase. We will include results in the final version to demonstrate the robustness and generalizability of our method.

---

> ### Comment · Reviewer_AwnR · 2024-11-26
> **Response to Authors**
>
> Thanks to the authors for their detailed response. The new experiments provide stronger evidence for the effectiveness of the two proposed losses. If a rating of 7 were available, I would raise my score to 7.

---

> ### Author Response · Authors · 2024-11-26
> **Thank you to Reivewer AwnR!**
>
> We are delighted to have addressed your concerns and appreciate your willingness to increase your score to “7”. Wishing you the very best with your own ICLR submission and all future submissions! Thank you once again for your valuable feedback, which has strengthened our work, and for your contribution to ICLR!
>
> Sincerely,
>
> The Authors

---

### Meta-Review · Area_Chair_8teq · 2024-12-11

**Metareview:**

This paper proposes that backdoor poisoning attacks on VLMs should be performed using out-of-distribution (OOD) samples, as the attacker may not have access to the training data distribution. In this context, the attacker still has access to the trained VLM and can fine-tune the model on poisoned OOD samples to inject a backdoor. This is accomplished using the proposed VLOOD method, which incorporates distillation-based loss functions. Reviewers raised concerns regarding the design of the loss functions and the weight adjustment mechanism. The authors addressed these concerns in detail and provided compelling new experimental results. After considering all reviews and the author’s rebuttal, I recommend acceptance of this paper.

**Additional Comments On Reviewer Discussion:**

Three reviewers submitted their reviews for this paper, with two positive ratings (6 and 8) and one negative rating (5). The negative rating came from Reviewer Mff3, who has been silent during the rebuttal and discussion phases. I reviewed the concerns raised and the authors' responses, and found that most of the concerns have been adequately addressed.

---

### Decision · Program_Chairs · 2025-01-22

Accept (Poster)